# SPARSE ISO-FLOP TRANSFORMATIONS FOR MAXIMIZING TRAINING EFFICIENCY

## ABSTRACT

Recent studies have explored the application of weight sparsity to enhance the training efficiency of DNNs in terms of test accuracy w.r.t training FLOPs. These studies have focused on reducing training FLOPs, but training with sparse weights often results in accuracy degradation or necessitates prolonged training schedules to attain performance similar to the original dense models; making the actual training efficiency gains less evident. In contrast, our work emphasizes leveraging sparsity to increase accuracy while maintaining the same FLOPs as the dense model, thereby demonstrating improved training efficiency through higher accuracy. We introduce Sparse-IFT, a family of Sparse Iso-FLOP Transformations that serve as drop-in replacements for dense layers, enhancing their representational capacity and FLOP efficiency. Each transformation is parameterized by a single hyperparameter (i.e., sparsity level), offering a broader search space for identifying optimal sparse masks. Substituting dense layers with Sparse-IFT, without altering any training hyperparameters, yields substantial improvements across a range of computer vision and natural language processing tasks; ResNet-18 on ImageNet (+3.5%) and GPT-3 Small on WikiText-103 (-0.4 PPL), both matching larger dense models that use 2x or more FLOPs. To our knowledge, this is the first work to demonstrate the use of sparsity for improving the accuracy of dense models, all while maintaining consistent training FLOPs budgets via a simple set of sparse transformations.

## 1 INTRODUCTION

Increases in model size and training data have led to many breakthroughs in deep learning (e.g., AlexNet (Krizhevsky et al., 2012), ResNet (He et al., 2016), Transformers (Vaswani et al., 2017), GPT (Radford et al., 2018; 2019), AlphaGo (Silver et al., 2017), etc.). Consequently, computational and memory demands for training and deploying deep neural networks (DNNs) have surged dramatically. To enable the deployment of large models, multiple techniques (e.g., distillation (Hinton et al., 2015), quantization (Han et al., 2015a), pruning (Han et al., 2015b)) have been introduced to reduce inference FLOPs and memory requirements. While these techniques improve inference efficiency (test accuracy w.r.t inference FLOPs), the associated training costs are still prohibitive. In this work, we focus on improving the training efficiency (test-accuracy w.r.t training FLOPs) of DNNs.

Recent works (Evci et al., 2020; Jayakumar et al., 2020) have explored using weight sparsity to reduce the FLOPs spent in training. Frankle & Carbin (2018) demonstrate that sparse subnetworks (termed "lottery tickets") exist at initialization and can be trained to match the accuracy of their original dense network. Inspired by this result, various dynamic sparse training (DST) methods (Ma et al., 2022; Evci et al., 2020; Liu et al., 2021b; Jayakumar et al., 2020) attempt to find optimal sparse subnetworks within a training run. While these methods primarily aim to improve training efficiency by reaching dense accuracy with fewer FLOPs, they often perform worse than their dense baselines or rely on longer training schedules (up to 2-5× training iterations) to close the gap (Yuan et al., 2021; Tai et al., 2022; Liu et al., 2021a). As a result, these techniques can sometimes even require more FLOPs than training the dense model (Ma et al., 2022; Evci et al., 2020; Jayakumar et al., 2020). Our aim is to highlight our unique contribution in utilizing sparsity to enhance standard dense model accuracy, distinguishing our work from previous research. While past studies focused on pruning techniques to improve accuracy of pretrained dense models (Han et al., 2015b; Liu et al., 2017; Molchanov et al., 2017), our innovation lies in demonstrating sparsity's impact on accuracy when training from scratch within the same

training FLOP budget as dense models. Specifically, we introduce a family of Sparse Iso-FLOP Transformations (Sparse-IFT) that can be used as drop-in replacements for dense layers in DNNs.

These transformations increase the representational capacity of layers and facilitate the discovery of optimal sparse subnetworks without changing the layer's underlying training and inference FLOPs (i.e., Iso-FLOP). For example, making a layer wider but sparser increases dimensionality while still maintaining FLOPs due to sparsity. All Sparse-IFT members are parameterized by a single hyperparameter, the sparsity level. Figure 1 summarizes the ImageNet performance with ResNet models, where our Sparse Wide IFT variants significantly increase the accuracy of matching Iso-FLOP dense models. In particular, Sparse Wide ResNet-18 at 90% sparsity improves the top-1 accuracy from 70.9% to 74.4% (+3.5%), and outperforms a dense ResNet-34 (74.2%) while using 2x fewer FLOPs. We emphasize that these gains were obtained by replacing dense layers with transformations from the Sparse-IFT family and required no changes to training hyperparameters. The main contributions of our work are:

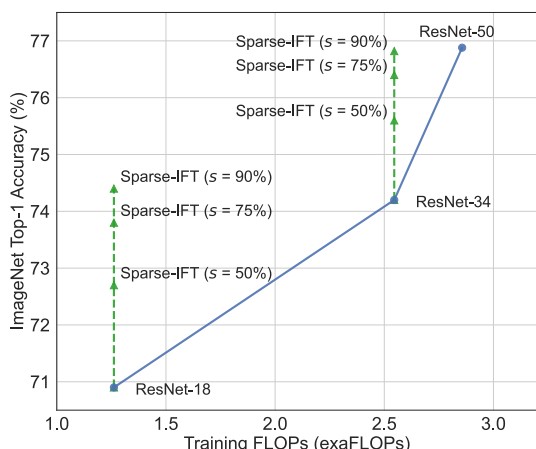

Figure 1: Accuracy vs. Training FLOPs for different variants of ResNet on ImageNet. Sparse-IFT provides significant accuracy gains across different models and sparsity levels while using the same FLOP budget as its dense counterpart.

1. We introduce Sparse Iso-FLOP Transformations (Sparse-IFTs), a family of techniques aimed at enhancing DNN training efficiency. These transformations boost accuracy while maintaining a constant FLOP count. Sparse-IFTs are characterized by a *single hyperparameter, sparsity level*, and can be seamlessly used as drop-in replacements for dense layers.

2. In the CV domain, using Sparse-IFT increases the top-1 accuracy of ResNet-18 and ResNet-34 by 3.5% and 2.6% respectively on ImageNet. Finetuning these pre-trained models for object detection (MS COCO) and segmentation (CityScapes) leads to an improvement of 5.2% mAP and 2.4% mIoU, respectively.

3. In the NLP domain, using Sparse-IFT with GPT-3 Small leads to a 0.4 perplexity improvement on the WikiText-103 language modeling task, and matches the PPL of a dense GPT-3 Medium while using 2.4x fewer training FLOPs.

## 2 METHOD

In this section, we present our method to improve training efficiency. We first explain our intuition and hypotheses, followed by our methodology.

### 2.1 TRAINING WITH DENSE MATRICES IS FLOP INEFFICIENT

Prior research indicates that modern DNNs are overparameterized, and they exhibit sparsity in both features and weights across layers. The Lottery Ticket Hypothesis (LTH) Frankle & Carbin (2018) demonstrates that sparse DNNs can achieve the same accuracy as dense counterparts when initialized with an effective sparsity mask ("lottery ticket"). These findings emphasize the advantage of sparse weight configurations over dense matrices during training. While sparse training methods are theoretically more efficient, their practical application often results in lower accuracy compared to dense baselines. This discrepancy may be attributed to the challenges of identifying "lottery tickets" within a single training run. While sparse models reduce the FLOPs needed per step, we hypothesize that existing sparse training methods make sub-optimal use of these computational savings. For example, state-of-the-art sparse training methods Jayakumar et al. (2020); Evci et al. (2020); Yuan et al. (2021); Tai et al. (2022); Liu et al. (2021a) invest these FLOP savings into longer training

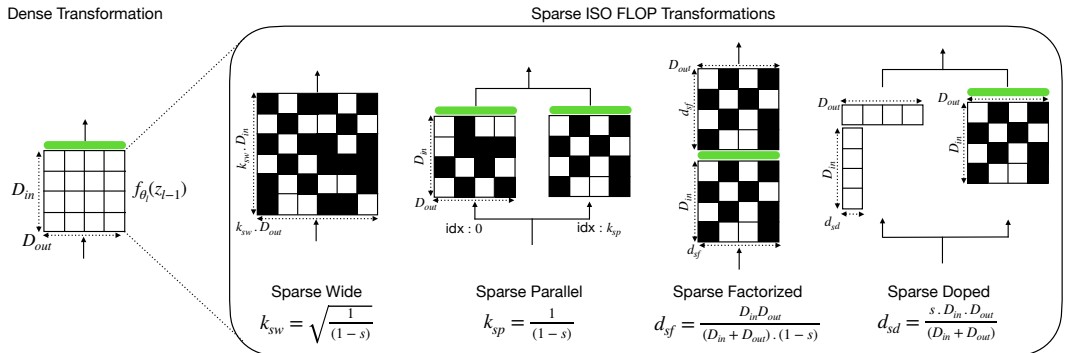

Figure 2: Different members of the Sparse-IFT family. Transformation of all members is parameterized by a single hyperparameter (i.e., sparsity level ($s$)). Black and white squares denote sparse and active weights, respectively. Green block indicates a non-linear activation function (e.g., BatchNorm, ReLU, LayerNorm). All transformations are derived with sparsity set to $50\%$ as an example, are Iso-FLOP to the dense feedforward function $f_{\theta_l}$, and hence can be used as a drop-in replacement of $f_{\theta_l}$. See Section 2.4 for more details about each member.

schedules to close the accuracy gap and compensate for the inability to discover an optimal mask earlier in training. This setup is inefficient since it ultimately requires more training FLOPs than the dense baseline to reach the same target accuracy. In our work, we take an orthogonal approach and invest these FLOP savings into (a) increasing the representational capacity of a layer and (b) increasing its search space, which we hypothesize can facilitate the discovery of an optimal sparse mask Ramanujan et al. (2020); Stosic & Stosic (2021). While utilizing larger sparsity-enabled models has exhibited accuracy improvement potential, the challenge lies in designing an appropriate architecture. For instance, when aiming to surpass the ResNet-18 performance on ImageNet, finding the right sparsity and larger network design is crucial. Many studies explore diverse combinations to balance sparsity and network size for outperforming dense models. However, these methods often lack FLOP efficiency, requiring multiple iterations for optimal settings and hyperparameter tuning. Therefore, we propose replacing dense transformations with FLOP-equivalent sparse transformations. We denote these transformations as the Sparse Iso-FLOP Transformation (Sparse-IFT) family.

## 2.2 SETUP

For clarity, we will explain our method for a fully connected neural network. In Appendix A.1, we detail the straightforward extension of our method to convolutional layers. Let $\mathcal{N}$ denote a $L$ layered DNN parameterized by $\Theta_{\mathcal{N}}$. Let $\Theta_{\mathcal{N}} \in \{\theta_1, ..., \theta_L\}$ denote the parameters of the DNN. The output of the $l$-th layer is defined as: $z_l = \sigma(f_{\theta_l}(z_{l-1}))$ for some activation function $\sigma$ (e.g., ReLU Nair & Hinton (2010)) and feedforward function $f_{\theta_l}$. Specifically, let $f_{\theta_l}(z_{l-1}) = \theta_l^T z_{l-1}$, where $\theta_l \in \mathbb{R}^{D_{in} \times D_{out}}, z_{l-1} \in \mathbb{R}^{D_{in} \times B}$ and $B, D_{in}, D_{out}$ denote the batch-size, input, and output dimensionality of features respectively. The total FLOPs needed for $f_{\theta_l}$ are given by $B \cdot D_{in} \cdot D_{out}$.

## 2.3 SPARSE ISO-FLOP TRANSFORMATIONS

In the standard setup, the feedforward function $f_{\theta_l}$ computes the output features as a linear transformation of input features. From a theoretical perspective, the feedforward function can make use of arbitrary non-linear transformations. However, in practice, most transformations are expressed as dense matrix multiplications due to widespread support on GPUs (Nvidia, 2023). As stated before, we are interested in improving the training efficiency of DNNs, by enhancing the representational capacity of the feedforward function. Naively increasing the representational capacity by stacking more layers Lin et al. (2014a), increasing width Zagoruyko & Komodakis (2016), mixture of experts Shazeer et al. (2016), etc. increases the computational FLOPs. In our work, we use unstructured sparsity in weight matrices and ensure that the FLOPs of the transformation are the same as that of a dense feedforward function. Let $\Psi_l$ denote the set of Sparse Iso-FLOP Transformations (Sparse-IFT) for a particular layer $l$:

$$\Psi_l : \{\psi_l(s), 0 \leq s < 1, g(\psi_l) \approx g(f_{\theta_l})\},$$

where $\psi_l$ is a transformation, $s$ represents the sparsity level, and $g(\cdot)$ returns the computational FLOPs. Each transformation in this set satisfies the following properties: (1) the computational FLOPs of the transformation $\psi_l$ are same as that of dense transformation $f_{\theta_l}$, and (2) the transformation is parameterized by a single hyperparameter - the sparsity level. Since these transformations are Iso-FLOP to the dense feedforward function, we can use them as drop-in replacements without affecting the FLOPs of a layer. While there may be other FLOP-invariant transformations, in this work, we detail four different members: Sparse Wide, Sparse Parallel, Sparse Factorized, and Sparse Doped.

## 2.4 MEMBERS OF SPARSE-IFT

**Sparse Wide** The sparse wide transformation augments the representational capacity of a layer by increasing the number of output features while keeping $s$ fraction of weights sparse. When using this transformation, we widen the input and output features for all the $L$ layers of the network with the same widening factor, $k_{sw}$, to avoid a mismatch in feature dimensionality across layers. Let $\theta_l^{sw} \in \mathbb{R}^{k_{sw} \cdot D_{in} \times k_{sw} \cdot D_{out}}$ denote the transformation matrix, with $s$ fraction of weights being sparse. Since the fraction of non-sparse weights is given by $1 - s$, the FLOPs required by this transformation are $B \cdot (k_{sw} \cdot D_{in}) \cdot (k_{sw} \cdot D_{out}) \cdot (1 - s)$. Setting these equal to the FLOPs of the original dense $f_{\theta_l}$, we obtain the widening factor $k_{sw} = \sqrt{\frac{1}{(1-s)}}$. If we set the sparsity $s$ to 0, we obtain $k_{sw}$ as 1 and recover the original dense feedforward function.

**Sparse Parallel** The sparse parallel transformation replaces the feedforward function with a sum of $k_{sp}$ non-linear functions. Let $\theta_l^{sp} \in \{\theta_l^{sp,1}, ..., \theta_l^{sp,k_{sp}}\}$ denote the parameters of this transformation, where $\theta_l^{sp,j} \in \mathbb{R}^{D_{in} \times D_{out}}$ denotes the transformation matrix of $j^{th}$ function, where $s$ fraction of weights are sparse. The sparse parallel transformation in this case is $\psi_l^{sp} = \sum_{j=1}^{k_{sp}} \sigma((\theta_l^{sp,j})^T z_l)$, where $\sigma$ is a non linear function. In practice, $\psi_l^{sp}$ is implemented as a layer with $k_{sp}$ parallel branches. The computational FLOPs of this transformation is $k_{sp} \cdot B \cdot D_{in} \cdot D_{out} \cdot (1 - s)$. Setting these FLOPs equal to FLOPs of $f_\theta$, we obtain $k_{sp} = \frac{1}{(1-s)}$. Note, at $s = 0$, the number of parallel branches $k_{sp}$ is 1. If we replace the non-linear function $\sigma$ with Identity, we can recover the original dense feedforward transformation.

**Sparse Factorized** The transformation matrix of the feedforward function $f_{\theta_l}$ is denoted by $\theta_l \in \mathbb{R}^{D_{in} \times D_{out}}$. Multiple works have explored matrix factorization techniques to express the transformation matrix $\theta_l$ as a product of two matrices $\theta_l = UV^T$, where $U \in \mathbb{R}^{D_{in} \times d}$, $V \in \mathbb{R}^{D_{out} \times d}$. Khodak et al. (2020); Tai et al. (2016) and Chen et al. (2021b) have explored low-rank factorization ($d << D_{out}$) as a form of structured sparsity to improve training and inference efficiency, while Arora et al. (2018) and Guo et al. (2020a) have explored overparameterized factorizations for better generalization and faster convergence. In contrast, we use factorization to augment the representational capacity without decreasing or increasing the FLOPs. More precisely, let $\theta_l^{sf} \in \{U_l, V_l\}$ denote the parameters of this transformation, where $U_l \in \mathbb{R}^{D_{in} \times d_{sf}}$, $V_l \in \mathbb{R}^{d_{sf} \times D_{out}}$ are sparse matrices with $s$ fraction of their weights being sparse. The functional transformation in this case is $\psi_l^{sf} = V_l^T \sigma(U_l^T z_l)$. The computational FLOPs of this transformation is $d_{sf} \cdot B \cdot (D_{in} + D_{out}) \cdot (1 - s)$. Setting these FLOPs equal to FLOPs of $f_{\theta_l}$, we obtain $d_{sf} = \frac{D_{in} \cdot D_{out}}{(D_{in} + D_{out}) \cdot (1-s)}$. Note, setting sparsity $s = 0$, we recover a non-linear low-rank factorization with dense matrices.

**Sparse Doped** family of transformation is inspired by works Chen et al. (2021a); Thakker et al. (2021); Udell & Townsend (2019); Candès et al. (2011) which approximate a dense matrix with a combination of low-rank factorization and sparse matrix. In our work, we replace the feedforward function with low-rank factorization (with rank $d_{sd}$) and an unstructured sparse weight matrix (with sparsity $s$). Let $U_l \in \mathbb{R}^{D_{in} \times d_{sd}}$, $V_l \in \mathbb{R}^{d_{sd} \times D_{out}}$ denote the low-rank matrices, and $\theta_l^{sd} \in \mathbb{R}^{D_{in} \times D_{out}}$ denote the matrix with unstructured sparsity. The functional transformation, in this case, is given by $\psi_l^{sd} = V_l^T(U_l^T z_l) + \sigma((\theta_l^{sd})^T z_l)$. The computational FLOPs associated with this transformation are $B \cdot d_{sd} \cdot (D_{in} + D_{out}) + (1 - s) \cdot B \cdot D_{in} \cdot D_{out}$. Setting these FLOPs equal to FLOPs of $f_{\theta_l}$, we obtain $d_{sd} = \frac{s \cdot D_{in} \cdot D_{out}}{(D_{in} + D_{out})}$. Note, as $s \to 0$ and $d_{sd} \to 0$, the low-rank component of the transformation disappears, and we can recover the dense feedforward function as a special case by setting $\sigma$ to Identity.

## 2.5 CARDINALITY OF SEARCH SPACE

One of our hypotheses is that increasing the search space of the sparsity mask via Sparse-IFT can make training more efficient. Results from past work support this hypothesis. Ramanujan et al. (2020) demonstrate that the odds of finding a lottery ticket in a randomly initialized network increase with the width of a network. Liu et al. (2022b) and Stosic & Stosic (2021) show that increasing the search space by increasing width or depth improves accuracy. In our work, we define the cardinality of a search space as the number of weights a sparse training method can explore. Table 1 characterizes the cardinality of search space for each member of the Sparse-IFT family.

The search space for Sparse Wide, Sparse Parallel, and Sparse Factorized transformations increase proportional to the width scaling factor, number of parallel branches, and size of intermediate hidden dimension, respectively. Sparse Doped transformation splits its computational FLOPs between low-rank factorization and unstructured sparse weight matrix. The size of the unstructured weight matrix is invariant to sparsity; thus cardinality of search space for this transformation is constant.

Table 1: Cardinality of search space of sparsity mask for different members of the Sparse-IFT family.

| Transformation | Cardinality of Search Space |
|---|---|
| Sparse Wide | $(k_{sw})^2 \cdot (D_{in} \cdot D_{out})$ |
| Sparse Parallel | $k_{sp} \cdot (D_{in} \cdot D_{out})$ |
| Sparse Factorized | $d_{sf} \cdot (D_{in} + D_{out})$ |
| Sparse Doped | $D_{in} \cdot D_{out}$ |

## 3 EXPERIMENTS

In this section, we demonstrate how transformations from the Sparse-IFT Family lead to improvements across a variety of different tasks in the CV and NLP domains. First, in Section 3.2, we describe the experimental setups and validate the design choices through multiple ablation studies on CIFAR-100 Krizhevsky et al. (2009), followed by results on ImageNet Krizhevsky et al. (2012). Then, in Section 3.5, we highlight the advantages of pre-training with Sparse-IFT through gains on downstream tasks. Next, we present the benefits of Sparse-IFT in the NLP domain by demonstrating results on GPT Brown et al. (2020) in Section 3.6. Unless stated otherwise, the results presented below are obtained by replacing all dense layers with a given transformation from the Sparse-IFT family while only tuning the sparsity level. All sparse models are trained using a uniform sparsity distribution (i.e., all layers have the same sparsity level). We adopt the default hyperparameters from RigL Evci et al. (2020) for dynamic sparsity. More details about the setup can be found in Appendix B.2.

### 3.1 IMPLEMENTATION DETAILS

**Computer Vision** We evaluate our method on CIFAR-100 and ImageNet using CNNs and hybrid Vision Transformer (ViT) networks. We follow published training settings for CIFAR-100 DeVries & Taylor (2017) and ImageNet Nvidia (2019b). For both datasets, we follow the standard evaluation procedures and report the top-1 accuracy. Details for model architectures, datasets, and training hyperparameters are given in Appendix B.2. All standard deviation was reported over 3 random seeds. However, for a select few computationally expensive experiments, we report results from a single run due to limited computational budget.

**Natural Language Processing** We evaluate Sparse-IFT by training GPT-3 Small (Brown et al., 2020) from scratch on the WikiText-103 (Merity et al., 2017) language modeling task, a commonly used NLP benchmark dataset. The compute cost and resources for training quickly become prohibitive when transforming GPT models with Sparse-IFT. Hence, we train our GPT models on the Cerebras CS-2 (Lie, 2022a;b) and leverage its ability to accelerate training with unstructured sparsity.

### 3.2 RESULTS AND ABLATIONS ON CIFAR-100

In this section, we conduct various ablations to validate our design choices. Unless stated otherwise, all experiments below are with ResNet-18 architecture on CIFAR-100.

**Importance of Dynamic Sparsity** All members of the Sparse-IFT family utilize transformations with unstructured sparsity. This study investigates the importance of the sparse training method when training different configurations of Sparse-IFT architectures. For this analysis, we focus on the Sparse Wide IFT and evaluate it with transformations obtained with sparsity $\in \{50\%, 75\%, 90\%\}$ using three

sparse training methods: static sparsity, SET Mocanu et al. (2018) and RigL Evci et al. (2020). RigL and SET are dynamic sparse training methods in which the sparsity mask evolves during training. The key difference is that RigL updates the mask based on gradient information, whereas SET updates the mask randomly. Results of our ablation are documented in Table 2. Here, the following trends can be observed: 1) the Sparse Wide IFT outperforms dense baselines across all operating points (sparsity and sparse training method), 2) dynamic sparse training methods (RigL and SET) obtain higher accuracies compared to training with static sparsity, and 3) gains with static sparsity plateau at lower levels of sparsity, while dynamic sparse training methods gain accuracy at higher sparsities. As mentioned in Section 2.5, Sparse-IFT transformations increase the search space $\propto$ sparsity. Dynamic sparse training methods can explore and exploit this increased search space Stosic & Stosic (2021) and therefore outperform training with static sparsity. Out of the two dynamic sparse training methods evaluated in our study, RigL consistently outperforms SET. Therefore, we use RigL as our sparse training method for all the experiments reported below.

Table 2: Sparse Wide IFT using various sparse training methods with ResNet-18 on CIFAR-100 across different levels of sparsity (columns). Best accuracy for each sparse training method is highlighted in bold.

| Dense | Sparse Method | 0.50 | 0.75 | 0.90 |
|---|---|---|---|---|
| | Static | **78.5 ± 0.3** | 78.3 ± 0.1 | 78.2 ± 0.3 |
| 77.0 ± 0.2 | SET | 78.8 ± 0.1 | 79.2 ± 0.2 | **79.8 ± 0.2** |
| | RigL | 79.1 ± 0.2 | 79.5 ± 0.1 | **80.1 ± 0.2** |

**Importance of Using Non-Linear Activations** Some members of the Sparse-IFT family are inspired by recent works which overparameterize the feedforward function during training and fold it back into a single dense matrix post training Ding et al. (2021b;a); Guo et al. (2020a); Ding et al. (2019). Although these works show the benefits of linear overparameterization, this comes at the cost of a significant increase in training FLOPs. In contrast, while we also increase the representational capacity of the feedforward function, we do so with an Iso-FLOP transformation. Since we remain Iso-FLOP to the original dense model, we do not require post-training modifications to collapse weight matrices for inference efficiency. This uniquely allows us to use non-linearities (e.g., ReLU) in members of the Sparse-IFT family to enhance the representational capacity of the network further. We validate the importance of this design choice by training ResNet-18 with Sparse Factorized IFT with and without non-linearities, and observe significant accuracy gains across all sparsity levels when using non-linear activations. For example, at 90% Sparse Factorized, using non-linearity, we see a 1.8% gain in test accuracy over the ResNet-18 CIFAR-100 dense baseline, compared to a drop of 0.5% without it. These findings hold for other members of the Sparse-IFT family as well (see Appendix B.1 for more details).

**Sparse-IFT ResNet-18** Here, we evaluate different members of the Sparse-IFT family on ResNet-18 and CIFAR-100 across different sparsity levels. Table 3 highlights the best accuracy achieved by each member of the Sparse-IFT family. Compared to the accuracy of the dense baseline (77%), all Sparse-IFT members obtain significant accuracy improvements using the same FLOPs as the dense model. We note that the Sparse Doped transformation is the only member of the Sparse-IFT family which does not gain accuracy at higher levels of sparsity. We hypothesize that this phenomenon occurs due to two reasons: (a) cardinality of the search space of the sparsity mask does not increase with sparsity level (see Table 1), and (b) the number of active weights in the unstructured matrix decreases $\propto$ sparsity. In Appendix B.3.1, we compare Sparse-IFT against other baselines obtained with sparse training methods (e.g., RigL and SET) under the same training efficiency setup. Specifically, we train ResNet-18 model on CIFAR-100 at sparsity levels $\in \{50\%, 75\%, 90\%\}$, and ensure that these runs use the same FLOPs as the dense baseline by

Table 3: Sparse-IFT families on CIFAR-100 with ResNet-18 model across different levels of sparsity (columns). Best accuracy of each transformation is highlighted in bold.

| Dense | Transformation | 0.50 | 0.75 | 0.90 |
|---|---|---|---|---|
| | Sparse Wide | 79.1 ± 0.2 | 79.5 ± 0.1 | **80.1 ± 0.2** |
| | Sparse Factorized | 77.8 ± 0.2 | 78.4 ± 0.5 | **78.9 ± 0.5** |
| 77.0 ± 0.2 | Sparse Parallel | 77.9 ± 0.4 | **79.1 ± 0.2** | 78.2 ± 0.2 |
| | Sparse Doped | **78.2 ± 0.1** | 77.8 ± 0.1 | 76.9 ± 0.2 |

extending the training iterations. Our results show that Sparse-IFT outperforms these competitive baselines by a significant margin.

**Sparse-IFT vs. Dense Overparametrization** The success of Sparse-IFT members can be attributed to efficient exploration of large search space with sparsity. Training this large search space in a dense manner leads to consumption of more training FLOPs than the dense baseline, but provides us with the upperbound (in terms of accuracy) for a sparse subnetwork. In this section, we will characetrize this gap between the Sparse-IFT members and their dense counterpart. In Table 4, we compare the sparse and dense counterparts of the two best performing Sparse-IFT members.

For both members, training in dense or sparse manner, leads to similar accuracy across all sparsity levels. This result demonstrates that training with sparsity allows for efficient exploration and exploitation of overparameterized space without incurring the computational cost of dense training of large neural

Table 4: Sparse-IFTs trained in a sparse and dense manner on CIFAR-100 with ResNet-18 for different levels of sparsity.

| Transformation | Train Method | 0.50 | 0.75 | 0.90 |
|---|---|---|---|---|
| Sparse Wide | Sparse | $79.1 \pm 0.2$ | $79.5 \pm 0.1$ | $\mathbf{80.1 \pm 0.2}$ |
| | Dense | $78.9 \pm 0.2$ | $79.7 \pm 0.1$ | $\mathbf{80.2 \pm 0.3}$ |
| Sparse Parallel | Sparse | $77.9 \pm 0.4$ | $\mathbf{79.1 \pm 0.2}$ | $78.2 \pm 0.2$ |
| | Dense | $78.1 \pm 0.2$ | $\mathbf{78.9 \pm 0.1}$ | $78.1 \pm 0.1$ |

networks. For example, dense runs (with transformations achieved with 90% sparsity) consume 10x more FLOPs compared to sparse runs.

**Unstructured vs. Structured Sparsity** We compare unstructured sparsity to structured sparsity with Sparse-IFT. In theory, for a fixed number of non-zero elements in a sparse mask, the use of unstructured sparsity can search over all the possible variations of the mask. However, since most hardware accelerators are not able to accelerate com-

Table 5: Sparse Wide IFT with unstructured and structured sparsity across different levels of sparsity (columns) on CIFAR-100 with ResNet-18.

| Dense | Sparsity Pattern | 0.50 | 0.75 | 0.90 |
|---|---|---|---|---|
| $77.0 \pm 0.2$ | Unstructured | 79.1 | 79.5 | **80.1** |
| | N:M Block Sparse | 77.1 | **78.4** | 78.1 |

putations with unstructured sparsity, multiple works have investigated training with structured sparsity (e.g., low-rank and block-sparse matrices) to obtain wall-clock speed-ups Khodak et al. (2020); Tai et al. (2016); Chen et al. (2021b); Hubara et al. (2021); Dao et al. (2022); Chen et al. (2022a). We study structured sparsity by deriving Iso-FLOP configurations using low-rank and block sparsity with Sparse Wide IFT. We use the method proposed in Hubara et al. (2021) to search N:M transposable sparsity, which can accelerate training on GPUs with Tensor Cores. In our evaluation, the low-rank factorization results were worse than block sparsity (see more details in Appendix B.3.3). Table 5 compares unstructured sparsity to block sparsity. Although using Sparse-IFT with block sparse matrices lead to improvements over the dense baseline, unstructured sparsity achieves the highest gains. This result can be explained by the fact that block-sparse matrices have reduced mask diversity (Hubara et al., 2021) compared to unstructured sparse matrices.

## 3.3 RESULTS WITH EFFICIENT ARCHITECTURES

To further understand the robustness of Sparse-IFT across different model families, we evaluate Sparse-IFT on architectures that are optimized for efficient inference (MobileNetV2 (Sandler et al., 2018) and MobileViT (Mehta & Rastegari, 2021)) and efficient training (BotNet (Srinivas et al., 2021)). We transform the dense layers in these architectures with Sparse

Table 6: Sparse Wide IFT with various efficient architectures on CIFAR-100 across different levels of sparsity (columns).

| Model | Dense | 0.50 | 0.75 |
|---|---|---|---|
| MobileNetV2 | $72.4 \pm 0.2$ | 73.4 | **73.7** |
| MobileViT-S | $73.5 \pm 0.1$ | 74.6 | **74.8** |
| BotNet-50 | $79.8 \pm 0.2$ | 80.3 | **80.6** |

Wide IFT and evaluate them at different sparsity levels. We observe a noticeable increase in test accuracy across all architectures (see Table 6). In addition, we demonstrate the robustness of the Sparse-IFT family by also applying the Sparse Parallel transformation and show consistent improvement across all architectures (see Appendix B.3.2). We evaluate the best performing architecture (BotNet-50) on ImageNet (see Section 3.4). The details of the experimental setup can be found in Appendix B.2.

## 3.4 RESULTS ON IMAGENET

We take the best performing Sparse-IFT transformations (i.e., Sparse Wide IFT and Sparse Parallel IFT) on CIFAR-100, and evaluate them on ImageNet using ResNet-18. Both families of Sparse-IFT obtain significantly higher accuracy compared to the dense baseline (refer to Table 7).

Table 7: Sparse-IFT on ImageNet. Best result for each transformation and architecture is highlighted in bold.

| Model | Dense | Transformation | Sparsity | | |
| | | | 0.50 | 0.75 | 0.90 |
|---|---|---|---|---|---|
| ResNet-18 | $70.9 \pm 0.1$ | Sparse Wide | 72.7 | 73.8 | **74.4** |
| | | Sparse Parallel | 72.7 | 73.2 | **74.0** |
| ResNet-34 | $74.2 \pm 0.1$ | Sparse Wide | 75.6 | 76.4 | **76.8** |
| BotNet-50 | $77.5 \pm 0.1$ | Sparse Wide | 77.9 | 78.3 | **78.5** |

Note, Sparse Wide IFT ResNet-18 at 90% sparsity improves over the dense baseline by 3.5%, and is able to match accuracy of dense ResNet-34 with $2\times$ fewer training FLOPs (see Figure 1). We take the best performing transformation (Sparse Wide IFT) and apply it to ResNet-34 and BotNet-50. Increasing sparsity leads to a consistent increase in accuracy, indicating improved training efficiency at higher sparsities. On BotNet-50, a hybrid ViT model, we see a 1% improvement at 90% sparsity.

## 3.5 TRANSFER LEARNING WITH SPARSE-IFT

To show the effectiveness of pre-training our Sparse-IFT classification backbones, we evaluate them on 1) object detection on MS COCO 2017 Lin et al. (2014b), and 2) semantic segmentation on CityScapes Cordts et al. (2016). For object detection, we adopt the RetinaNet Lin et al. (2017b) framework from the MMDetection open-source toolbox Chen et al. (2019) and report results in the standardized training setting. For semantic segmentation, we utilize DeepLabV3+ Chen et al. (2018) in the MMSegmenation open-source toolbox Contributors (2020). We evaluate ResNet-18 with Sparse Wide IFT (best-

Table 8: Sparse-IFT variants of ResNet-18 as backbones on downstream tasks : (a) Object detection on MS COCO, (b) Semantic segmentation on Cityscapes.

| | Metric | Dense | Sparsity | | |
| | | | 0.50 | 0.75 | 0.90 |
|---|---|---|---|---|---|
| MS COCO | AP | 29.3 | 31.3 | 32.8 | **34.5** |
| | $AP_{50}$ | 46.2 | 49.0 | 51.0 | **53.5** |
| | $AP_{75}$ | 30.9 | 33.0 | 34.8 | **36.5** |
| CityScapes | mIoU | 76.7 | 77.9 | 78.9 | **79.1** |
| | mAcc | 84.4 | 85.1 | 85.7 | **86.0** |

performing transformation on ImageNet). To ensure FLOP-equivalent comparisons with the dense backbone, the Sparse-IFT backbones remain sparse during fine-tuning. Appendix B.3.4 provides more details regarding the training setup. We summarize our findings in Table 8, where using Sparse Wide IFT ResNet-18 backbone leads to significant accuracy gains across all metrics on both downstream tasks.

## 3.6 RESULTS ON GPT END-TO-END TRAINING

We train the Sparse Wide IFT GPT-3 Small models at 50% and 75% sparsity levels, and compare against the standard dense GPT-3 Small and GPT-3 Medium models. Following Dao et al. (2022), we train all models from scratch on the WikiText-103 dataset and report the average test perplexity (PPL) over 3 random seeds in Table 9. We show that Sparse Wide IFT GPT-3 Small at 50% sparsity improves the perplexity

Table 9: Sparse-IFT for pre-training GPT-3 Small from scratch on WikiText-103 and report the test perplexity (lower is better).

| | Dense | 0.50 | 0.75 |
|---|---|---|---|
| GPT-3 Small | $20.8 \pm 0.3$ | **20.4** | 22.1 |

by 0.4 over its dense counterpart. We also note that the Sparse Wide IFT GPT-3 Small model performs comparable to a dense GPT-3 Medium ($20.5 \pm 0.2$ PPL) while using 2.4x fewer training FLOPs. In Appendix C.1, we provide details on the hyperparameters and how the total training FLOPs for the models in Table 9 were calculated.

**GPT Pre-training and Fine-tuning** While not the primary focus of our method, we note that Sparse-IFT can also be applied in a fine-tuning setup for NLP models. After pre-training sparse, the Sparse-IFT model can be fine-tuned as-is (i.e., remains sparse) or after densifying (i.e., allow the zeroed weights to learn) using a technique such as SPDF (Thangarasa et al., 2023). We perform some preliminary fine-tuning studies on BERT and GPT and those results can be found in Appendix C.2.

## 4 RELATED WORK

Our work is similar to the body of work studying the role of overparameterization and sparsity for training DNNs. The modeling capacity needed to learn a task is often unknown. Hence, we often solve this by training overparameterized models to fully exploit the learning capability and then compress them into a smaller subnetwork.

**Overparameterization**    Nakkiran et al. (2021) show that DNNs benefit from overparameterization. Following this, there have been many works that leverage overparameterization by scaling the size of models Rae et al. (2021); Goyal et al. (2022) and augmenting existing DNNs to increase modeling capacity and the accuracy of trained networks Guo et al. (2020b); Ding et al. (2019; 2021b); Cao et al. (2022); Vasu et al. (2022); Liu et al. (2022a). These methods use linear parameterizations of the model, making them highly inefficient to train, and are focused on improving inference throughput (reduced latency). In contrast, our work is focused on improving the modeling capacity using sparse non-linear parameterizations, which do not increase training FLOPs compared to the baseline model. While both approaches have the same inference FLOPs, our approach improves accuracy without increasing the training FLOPs.

**Sparse Training**    LTH (Frankle & Carbin, 2018; Frankle et al., 2020) shows that accurate sparse subnetworks exist in overparameterized dense networks but require training a dense baseline to find. Other approaches have proposed frameworks for identifying lottery tickets (Zhou et al., 2019; Ma et al., 2022) but still require a lot of compute resources. Following this, various attempts have been made to find the optimal sparse subnetwork in a single training run. These methods either try to find the subnetworks at initialization Tanaka et al. (2020); Wang et al. (2020a); de Jorge et al. (2020); Lee et al. (2018) or dynamically during training Mocanu et al. (2018); Evci et al. (2020); Jayakumar et al. (2020); Raihan & Aamodt (2020). However, given a fixed model capacity, these methods tradeoff accuracy relative to the dense baseline to save training FLOPs. Stosic & Stosic (2021) and Ramanujan et al. (2020) increase the search space during sparse training to retain accuracy; however, do not guarantee FLOPs savings. In contrast to these methods, our work introduces a set of non-linear sparse transformations, which increase the representational capacity of the network. Our approach does not entail the introduction of a novel sparse training algorithm. Instead, it enhances the search space of existing methods, resulting in improved generalization without compromising training efficiency.

**Iso-Parameter vs. Iso-FLOP**    Recent sparsity literature is focused on improving generalization at high sparsity levels. Hence, layer-wise sparsity distributions such as the Erdös-Rényi-Kernel Evci et al. (2020), Ideal Gas Quota Chen et al. (2022b), and parameter leveling Golubeva et al. (2021) are often used with sparse training to boost accuracies. However, these works target the setting where the models being compared have a fixed parameter budget (i.e., Iso-Parameter), which does not translate to similar training FLOPs to the original dense model (especially in CNNs). As a result, training models with these distributions often require different memory or computational resources per layer. Our approach does not focus on this Iso-Parameter setting but instead adopts the uniform sparsity distribution (i.e., every layer gets the same sparsity level), ensuring uniform FLOP reductions across the network. We also ensure the same computational FLOPs of a dense network by leveraging sparsity along with our Iso-FLOP transformations.

## 5 CONCLUSION

We introduce a new family of Sparse Iso-FLOP Transformations (Sparse-IFT) to improve the training efficiency of DNNs. These transformations can be used as drop-in replacements for dense layers and increase the representational capacity while using sparsity to maintain training FLOPs. This increase in capacity also translates to a larger search space allowing sparse training methods to explore better and identify optimal sparse subnetworks. For the same computational cost as the original dense model, Sparse-IFT improves the training efficiency (test accuracy w.r.t training FLOPS) across multiple model families in the CV and NLP domains for various tasks. We hope our work will open new investigations into improving the accuracy of DNNs by leveraging sparsity, particularly in light of advancements in hardware accelerators that offer improved support for weight sparsity during the training process.

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
