# OpenReview forum: "Sparse Iso-FLOP Transformations for Maximizing Training Efficiency"
_ICLR.cc/2024/Conference — ICLR 2024 Conference Withdrawn Submission_

### Official Review · Reviewer_aB9B · 2023-10-26

**Soundness:** 3 good
**Presentation:** 3 good
**Contribution:** 2 fair
**Rating:** 5
**Confidence:** 4

**Summary:**

This work proposes a family of so called ISO-FLOP transformations, where the base dense architecture is replaced by a wider sparse network with the same amount of FLOPs. Several sparsity patterns are considered, and the introduced approach is validated on several neural network architectures and tasks.

**Strengths:**

The proposed Sparse-IFT method attains pretty strong performance on several tasks and significantly improves upon the dense model given the same number of FLOPs.

Authors consider several distinct sparsity patterns with very different structure and properties and validate their method on multiple tasks and families of neural network architectures.

**Weaknesses:**

While the experimental results are quite strong, the work itself lacks scientific novelty and insights.
Sparsity models appear to improve upon dense models in terms of performance / FLOPs tradeoff, but the reason or at least intuition behind the advantage of sparse model is not clear. I think the work could greatly benefit if one showed that sparse models are more expressive than dense or have faster convergence. Even on a toy example.

Sparsity patterns in this work are adopted from the prior work and cannot be considered as novel.

**Questions:**

What is the relative latency of the sparsity patterns considered compared to dense model with the same number of parameters. There exist dedicated sparsity engines that are capable of accelerating sparse matrix operations, i.e. DeepSparse [1] for CPU and Sputnik [2], [3] for GPU. Can one expect that at least for some sparsity levels considered (i.e ~50%) throughput of the sparse model would be only marginally slower?

---
[1] https://github.com/neuralmagic/deepsparse

[2] https://github.com/google-research/sputnik

[3] https://github.com/microsoft/SparTA

---

### Official Review · Reviewer_Vqzr · 2023-11-01

**Soundness:** 4 excellent
**Presentation:** 4 excellent
**Contribution:** 3 good
**Rating:** 5
**Confidence:** 4

**Summary:**

This paper studies sparse training techniques with the goal of enhancing accuracy while preserving the training FLOPs at the same level as dense models. To accomplish this, this paper introduce a family of Sparse-Iso-FLOP Transformations (Sparse-IFT),  which replace the dense layers with sparse  transformations, including sparse wide, sparse parallel, sparse factorized, and sparse doped. A wide range of experiments in the domains of both vision and language is conducted, providing empirical evidence that underscores the efficacy of the proposed Sparse-IFT approach.

**Strengths:**

1. The concept of preserving the training FLOPs of a dense model while enhancing its representational capacity is both innovative and interesting. In my view, this direction holds significant promise as a burgeoning research avenue within sparse networks, with the potential to push the limits of current state-of-the-art dense models.


2. While this paper presents a range of methods for introducing sparsity, it simplifies the user experience by requiring only a single hyperparameter, the sparsity level, to adjust.

3. This paper performs comprehensive experiments across a spectrum of vision and language tasks in both pre-training and transfer learning scenarios, providing compelling evidence for the effectiveness of the proposed Sparse-IFT.

**Weaknesses:**

1. This paper lacks some crucial ablation studies. While it asserts that, given a fixed FLOPs budget, we can enhance the accuracy of the current network capacity with Sparse-IFO, it's important to note an alternative approach, which involves pruning a larger dense model to the same FLOPs level and then training it with sparse training techniques. For instance, when considering the task of training a dense ResNet-18 on ImageNet within a specific FLOPs budget, this paper showcases how accuracy can be increased from 70.9% to 74.4% at a sparsity level of 0.9 using Sparse-IFT. However, an alternative strategy would be to prune a ResNet-50 or a Wide ResNet-18 to a certain sparsity level (e.g., maybe 80% sparsity) such that these pruned networks share the same FLOPs as the dense ResNet-18. Subsequently, we apply sparse training techniques to train this pruned networks. The comparison between the accuracy of the pruned ResNet-50/Wide-ResNet-18 and the scaled ResNet-18 could provide valuable insights, which could help the community understand both direction better.


2. A more in-depth exploration of GPU memory usage are preferred. The methods introduced in this paper, such as Sparse Wide, Sparse Parallel, and Sparse Factorized, have the capability to effectively double the dimensions of weight matrices, potentially increasing GPU memory demands during training. Thus, it is better to provide an analysis that elucidates the trade-off between memory utilization and varying sparsity levels.

3. There appears to be a pattern where the proposed method consistently yields more significant improvements on relatively smaller networks. For instance, when examining Table 7, at a sparsity level of 90%, we observe a substantial 3.5% improvement on ResNet-18 but only a modest 1.0% enhancement on BotNet-50. This raises the question of whether these findings hold true for larger and deeper networks. Ideally, the focus should be on demonstrating that increasing model capacity can push the limit of state-of-the-art models, rather than solely improving the accuracy of smaller models.

**Questions:**

Please address the above weaknesses, especially 1 & 2 in the rebuttal.

---

### Official Review · Reviewer_Azx4 · 2023-11-04

**Soundness:** 3 good
**Presentation:** 3 good
**Contribution:** 2 fair
**Rating:** 3
**Confidence:** 4

**Summary:**

This paper introduces a new family of Sparse Iso-FLOP Transformations (Sparse-IFT) to improve the training efficiency of DNNs. The authors conduct extensive experiments to validate their ideas. For the same computational cost as the original dense model, Sparse-IFT improves the training efficiency across multiple model families in the CV and NLP domains for various tasks.

**Strengths:**

1. The experiment section is sound and the paper's presentation is good.
3. The proposed four FLOP-invariant transformations look brilliant.

**Weaknesses:**

My major concerns include three aspects:
1. The paper's contribution is limited, the only novel components are the four FLOP-invariant transformations. However, a similar idea of improving exploration in sparse training processes has been considered in existing works, e.g. [1][2].

[1] Jayakumar, Siddhant, Razvan Pascanu, Jack Rae, Simon Osindero, and Erich Elsen. "Top-kast: Top-k always sparse training." Advances in Neural Information Processing Systems 33 (2020): 20744-20754.

[2] Huang, Shaoyi, Bowen Lei, Dongkuan Xu, Hongwu Peng, Yue Sun, Mimi Xie, and Caiwen Ding. "Dynamic sparse training via balancing the exploration-exploitation trade-off." In 2023 60th ACM/IEEE Design Automation Conference (DAC), pp. 1-6. IEEE, 2023.

2. Although the authors claim training with sparse weights often results in accuracy degradation or necessitates prolonged training schedules to attain performance similar to the original dense models, the experiments of the paper are mainly focused on unstructured sparsity, which means the exact wallclock running time will be longer than dense models. Even the authors also conduct experiments on structured N:M sparsity. However, to the best of my knowledge, currently, the Nvidia Ampere GPU only supports acceleration on static 2:4 sparsity.

3. Besides, the performance of the model equipped with Sparse Iso-FLOP Transformations improvement compared to the dense model is also marginal.

**Questions:**

1. What is the exact wall clock running time of each experiment?
2. Apart from sparse training methods, have you considered experiments on pre-training sparsification methods, e.g. SNIP [1]?

[1] Lee, Namhoon, Thalaiyasingam Ajanthan, and Philip HS Torr. "Snip: Single-shot network pruning based on connection sensitivity." arXiv preprint arXiv:1810.02340 (2018).

3. From Table 9, the maximum sparsity level of GPT-3 to maintain dense performance is only 50% sparsity, which is much lower than those CV models. Can you comment on this fact?

4. In Table 2, static, SET, and RigL with the proposed sparse transformations are compared. However, can the authors compare the proposed sparse transformation methods with original weight sparsification patterns in SET or RigL?

---

### Official Review · Reviewer_WRvo · 2023-11-04

**Soundness:** 2 fair
**Presentation:** 3 good
**Contribution:** 2 fair
**Rating:** 3
**Confidence:** 3

**Summary:**

This paper proposes a family of techniques to transform a dense neural network into another neural network. The transformed model is larger in size (e.g., have a larger width), but has a similar inference FLOPs due to its parameter sparsity. Among various options, spanning from overparameterization via LoRa-like doping to doubling the depth, simply widening the width and adding sparsity works consistently well. The experimental results show that the transformed 'iso-FLOP' model tend to achieve a better accuracy than dense models.

**Strengths:**

- It is definitely a very interesting observation that widening the model via "sparse-wide" works better than other options (parallel, factorized, doped). This suggests that the classic technique of network pruning may indeed be a better choice than other methods to reduce the parameters of a dense neural network.
- The empirical evaluations have taken place on diverse, (relatively) up-to-date set of network architectures, such as GPT-3, MobileViT, BotNet, etc.
- The breadth of empirical exploration is also rich. Table 2, which explores how various sparse training algorithms affect the final performance, and table 5, which explores the impact of imposing n:m sparsity, is interesting and informative.

**Weaknesses:**

Overall, it is still very difficult for me to fully grasp the benefit of the proposed method over the existing techniques. In particular, here are some concerns:

- **Novelty.** It is quite difficult for me to clearly tell apart the proposed 'sparse-wide' method from the standard approach of pruning the over-parameterized model (at initialization). In fact, the main message of this paper---having a wider, sparse model is better than smaller, dense model---has already been confirmed at the classic paper of "To prune or not to prune" by Zhu & Gupta (2017).
- **Baseline.** Related to the previous question---if authors indeed think that the proposed method is any different from random pruning or performing pruning-at-initialization to a widened model, I strongly suggest adding these as a baseline and comparing the performances of such methods. Currently, there is no baseline method in all the tables, so it is really difficult to know the difference.
- **Practical benefit.** I am not sure these claimed *iso-FLOP* will indeed have a similar computational footprint, in terms of wall-clock latency or memory requirement. First, the widened model may have more number of activations, requiring more memory space to be loaded and trained. Second, sparse models often need more memory space to load their sparse weights (e.g., in CSR), and loading these through limited memory bandwidth incurs a lot of delay and power consumption. Third, there is no satisfactory hardware that can handle sparse operations with the same speed as iso-FLOP dense operations. Thus, I am not really sure if these can be really considered iso-computation.
- **Training Speed Comparison.** Sparse models are known to require more number of SGD steps to converge to the minimum (see, e.g., RigL). I wonder if this is true for the proposed iso-FLOP model.

**Questions:**

Most of my questions are in the "weaknesses" section. Plus, I have one more.
- **Initialization.** I think I missed this part, but how are the parameters for the transformed networks initialized?